# Expert Demand for Consumer Sleep Technology Features and Wearable Devices: A Case Study

**Jaime K Devine** [1,*], **Lindsay P. Schwartz** [1], **Jake Choynowski** [1] **and Steven R Hursh** [1,2]

1   Institutes for Behavior Resources, Baltimore, MD 21218, USA; lpschwartz@ibrinc.org (L.P.S.); jchoynowski@ibrinc.org (J.C.); shursh@ibrinc.org (S.R.H.)
2   Department of Psychiatry and Behavioral Sciences, Johns Hopkins University School of Medicine, Baltimore, MD 21205, USA
*   Correspondence: jdevine@ibrinc.org; Tel.: +1-410-752-6080 (ext. 132)

**Abstract:** Global demand for sleep-tracking wearables, or consumer sleep technologies (CSTs), is steadily increasing. CST marketing campaigns often advertise the scientific merit of devices, but these claims may not align with consensus opinion from sleep research experts. Consensus opinion about CST features has not previously been established in a cohort of sleep researchers. This case study reports the results of the first survey of experts in real-world sleep research and a hypothetical purchase task (HPT) to establish economic valuation for devices with different features by price. Forty-six (N = 46) respondents with an average of 10 ± 6 years' experience conducting research in real-world settings completed the online survey. Total sleep time was ranked as the most important measure of sleep, followed by objective sleep quality, while sleep architecture/depth and diagnostic information were ranked as least important. A total of 52% of experts preferred wrist-worn devices that could reliably determine sleep episodes as short as 20 min. The economic value was greater for hypothetical devices with a longer battery life. These data set a precedent for determining how scientific merit impacts the potential market value of a CST. This is the first known attempt to establish a consensus opinion or an economic valuation for scientifically desirable CST features and metrics using expert elicitation.

**Keywords:** behavioral economics; wearables; consumer sleep technology; Internet of Things; economical survey; expert elicitation

## 1. Introduction

Wearables are a hallmark of the Internet of Things (IoT), but sleep researchers have been using a precursor to wrist-worn wearables, called actigraphy, since before the "birth" of the internet in 1983 [1–3]. Internet-enabled applications and wearables began to hit the market in the mid-2010s [1,4]. A 2015 *Journal of Clinical Sleep Medicine* article coined the term used to describe publicly available, computer-based systems that aimed to monitor or improve individual sleep behavior, defined as "consumer sleep technologies (CSTs)" [5]. This term is widely used by the sleep research community to describe wearable or non-wearable sleep tracking technology [1]. As the name suggests, CSTs have been designed for the everyday consumer, rather than as a reliable scientific tool. However, CSTs are increasingly becoming a part of the research landscape [1,6–14].

Enough reviews, statements, editorials, validation studies, and research findings focused on CSTs have been published that it is impossible to cite them all, but none of these previous works have put a price on the value of CST features that the scientific community considers to have merit for clinical or research purposes (i.e., their scientific merit). Given that recent advancements in sleep-tracking technology have been driven by consumer economic demand, the time has come to examine researcher preferences in economic terms. Multiple publications have discussed the value of the measurements that CSTs collect for

use in a clinical or research setting [7–15], but the scientific merit of device features has not previously been examined in the context of behavioral economics. Behavioral economics is a field of science that applies behavioral science within economic frameworks to explain decision-making behavior [15–17]. The motivation for this project is to describe the value of scientifically meritorious CSTs in quantifiable terms, i.e., United States dollars (USD), that can then be used to guide the development of future CSTs in order to maximize scientific, as well as economic, gains.

This paper presents the results of an expert survey eliciting the professional opinions and economic valuation of CSTs from a sample of researchers interested in real-world sleep outside a controlled laboratory environment. To our knowledge, this is the first assessment of its type, and it is also the first report in an on-going, multi-step project conducted as a collaboration between the Institutes for Behavior Resources (IBR) Operational Fatigue and Performance group and the Behavioral Economics group to establish the economic value of CST wearable features on the basis of scientific merit. In order to assess the economic value of CST features that are desirable to academic and industry sleep researchers who are well-educated about the value of sleep monitoring, we first needed to establish which features are, in fact, desirable to sleep researchers in economic terms. This case study was designed to answer the following specific questions:

1. Which metrics of sleep quantity and quality do experts in the field believe are most important for a CST wearable to measure?
2. What wearable design features are most important for the successful tracking of sleep in the real-world from the perspective of experts who conduct such studies?
3. How much economic value do experts place on a CST wearable that has the most desirable sleep metrics and design features?

The first step in this project was to conduct an online survey to see what wearable features scientists with a professional interest in measuring real-world sleep consider most important [18]. Survey items were geared toward identifying discrete design features, as well as a hypothetical purchase task (HPT) and a validated behavioral economic demand procedure for evaluating demand across a range of hypothetical circumstances [19–21]. This procedure can provide information about sensitivity to price for a particular device or feature (e.g., how purchasing behavior may change as a function of cost) and provide an economic valuation of those features in terms of demand elasticity. The target recruitment population for this survey was scientists and industry professionals who routinely conduct research related to human sleep physiology or behavior in real-world environments, i.e., sleep research experts.

One concern with an expert survey is that the sample population may not be adequately representative of the expert community [22]. In qualitative research, even studies presenting the results from a single case can be highly informative; however, it is necessary for qualitative researchers to outline and justify their final sample size [23]. Adequate sampling requires the determination of a sufficient sample size based on an estimation of the size of the overall population [24]. Experts can be recruited using nonprobability sampling techniques, such as convenience sampling [25,26] or purposeful sampling [27–29]. "Snowball" sampling, in which participants identify other potential participants, is another nonprobable sampling technique which can help increase the recruitment of experts for survey participation [22,30,31]. This paper describes the efforts taken to ensure that respondents were representative of the global sleep expert population and presents a rank ordering of researcher preferences for CST device features and metrics.

The motivation of this case study has been to establish experts' consensus opinion of the economic value of desirable CST wearable features for use in real-world sleep research environments. The novelty of this study is that this is the first report of an examination of preferred CST wearables features solicited from sleep experts and a behavioral economic analysis of the value this group assigns to those features. The goal of this line of research is to bridge a communication gap that currently exists between CST manufacturers and the scientific community. Findings from this case study and subsequent surveys in this

on-going project can be used to assess the extent to which currently available devices meet scientific criteria and how general consumer demand for CST wearables is influenced by scientifically meritorious design. The significance of quantifying demand for CSTs by price based on scientific merit is to provide developers with incentive to improve their technologies in a manner that also benefits the sleep research and medical community.

## 2. Materials and Methods

This study was approved by the Salus Institutional Review Board and these analyses were conducted in accordance with the Declaration of Helsinki. Professional opinions from sleep medicine experts were elicited to identify what metrics and device features for measuring sleep outside the laboratory are most desirable to the scientific community. Potential respondents were recruited actively through direct contact and passively through social media on the Twitter (www.twitter.com (accessed on 13 March 2022)) and LinkedIn (www.linkedin.com (accessed on 13 March 2022)) social media platforms and through scientific presentations which described the scope of the project to the target audience. Potential respondents were actively recruited through email based on the results of a literature review conducted using the biomedical literature search engine Pubmed.gov (https://pubmed.ncbi.nlm.nih.gov/ (accessed on 13 March 2022)) and scholarly literature web search engine Google Scholar (https://scholar.google.com/ (accessed on 13 March 2022)).

The search criteria for the literature review included that the article had been peer-reviewed, published before July 2021, and included a combination of the terms 'sleep,' and 'environment' or 'operational' or 'real-world' or 'ecological' and/or 'wearable' or 'consumer sleep technology' or 'device' in the topic or title fields. Each returned article was manually scanned for relevancy. Articles which included a description of methods of active data collection in a real-world environment or simulated real-world environment were considered relevant. Review articles, meta-analyses, papers focused on device engineering/development, or opinion pieces such as editorials were not sufficient for inclusion. Authors who had published at least two or more relevant papers were subsequently considered subject matter experts and were contacted via blind carbon copy (bcc) email with a brief written explanation of the purpose of the research and a link to the online survey. Recruitment terminology on all platforms (social media, scientific presentation, and email) included a request to share the survey with any interested colleagues in relevant fields in order to create a snowball effect and reach a broader range of potential respondents.

The survey was hosted through the online tool Qualtrics (www.qualtrics.com (accessed on 13 March 2022)) between April and July 2021. The voluntary anonymous survey was composed of 42 questions grouped in 5 sections. The first section contained 5 questions that focused on identifying each respondent's background and research experience. The second section contained 10 questions about respondents' typical sample population and study design. The third section contained 13 questions asking about device preferences, and the fourth section asked respondents to rate their agreement with 6 statements. The fifth section contained 9 questions regarding economic demand for devices with varying features and price points. Respondents were able to provide comments in a text box with a 20k character limit at the end of the survey.

An annotated list of survey items and hypothetical devices for discussion in this paper is outlined in Table 1. Respondents could select the best fitting option from a multiple-choice list for Q1–Q12 or provide a write-in response if no option described them. Respondents were asked to rank by order of importance (high importance, medium importance, or low importance) a list of features or metrics related to the question topic for Q13–Q15. Respondents could additionally provide and rank a write-in response.

**Table 1.** Survey questions and number of responses.

| Question | Number of Responses |
|---|---|
| **Research Background and Experience** | |
| Q1. Do you conduct human subjects research related to sleep in real-world environments/outside a controlled laboratory environment? (*Only respondents who selected yes were able to complete the rest of the questions*) | 55 |
| Yes | 46 |
| Q2. How many years' experience do you have conducting human sleep research in real-world environments? | 44 |
| Q3. Which organization best describes your research affiliation? | 43 |
| Q4. In what region are you/your research based? | 42 |
| Q5. Which category best describes the population whose sleep you study? | 40 |
| Q6. Which category best describes the focus of your research? | 36 |
| **Device Preferences: Multiple Choice** | |
| Q7. Where is your preferred placement for a fieldable device or instrument to collect sleep data? | 32 |
| Q8. What do you consider to be the most important time scale for measuring sleep for your research in general? | |
| Q9. What is the most appropriate method for determining actual sleep onset/offset in real-world environments? | 33 |
| Q10. Do you collect data related to napping or fragmented sleep? (yes/no) | 29 |
| Yes | 24 |
| Q11. What is the most appropriate minimum period of inactivity that could reliably be considered a nap? (only respondents who selected Yes on Q10 received this question) | 23 |
| Q12. What is your preferred continuous observation period or window for collecting data on real-world sleep? | 32 |
| **Device Preferences: Rank Order of Importance** | |
| Q13. Which information about sleep do you consider most important to your research? | 33 |
| Q14. Which features of a fieldable device or instrument are/would be most important to facilitating data collection for your research? | 30 |
| Q15. Which factors related to devices or instruments are most important to limiting your observation period/data collection window? | 33 |

All data were exported from Qualtrics as an Excel file and subsequently analyzed using Excel 2013. In-depth statistical testing was not appropriate for these analyses due to the nature of the survey and the small sample size. Responses for Q13–Q15 were weighted by level of importance. The Excel Rank function was used to calculate weighted mean rank order for Q13–Q15 items. The hypothetical CST devices used for the HPT are briefly described in Table 2. A total of 18 participants responded to the HPT section of the survey.

Participants' data were first analyzed with algorithms used to determine the non-systematic demand curve data [21]. Four demand curves were excluded as the data were only input for a single price (one curve for Device A and B and two curves for Device C). The demand curve data were analyzed using a recent extension of the exponential demand model [32] specifically designed to handle zero demand data using guidelines and equations from Gilroy et al. (2021) [33]. With this equation, $Q_0$ is an estimate of the maximum level of demand (the number of devices one would purchase) and $\alpha$ is an estimate of the rate of change in elasticity normalized to the transformed maximum level of demand ($Q_0$). A demand curve template for GraphPad Prism 8.0 available from the Institutes for Behavior Resources (www.ibrinc.org (accessed on 13 March 2022)) was used to fit the pooled consumption data and estimate the two parameters. Post hoc extra sum of squares F-tests were used on the pooled consumption data to determine whether the demand elasticity rate parameter ($\alpha$) differed between devices. We also report the essential value (EV), which

is proportional to the inverse of $\alpha$ (EV = $1/(100 \times \alpha)$) and allows for a simple way to understand the rate of change in elasticity—a lower rate of change in elasticity is denoted by smaller $\alpha$ values and indicates a higher EV or higher resistance to the effect of price [32]. Taking the inverse of $\alpha$ ensures that a higher value is represented by a higher number. P*max* was also calculated for each curve. P*max* denotes the price at which demand becomes elastic and monetary expenditure would be maximal, and it was calculated using an Excel solver tool that uses $\alpha$ and $Q_0$ to estimate P*max* (Behavioral Economics Tools. Available online: https://ibrinc.org/behavioral-economics-tools/ (accessed on 14 March 2022)).

**Table 2.** Device descriptions for the hypothetical purchase task.

| | Device A | Device B | Device C |
|---|---|---|---|
| **Specifications** | <ul><li>Accelerometer</li><li>Light exposure</li><li>30 s epochs</li><li>30 days continuous battery life</li></ul> | <ul><li>Accelerometer</li><li>Light exposure</li><li>Heart rate detection</li><li>Oxygen saturation</li><li>4 days continuous battery life</li></ul> | <ul><li>Accelerometer</li><li>Light exposure</li><li>Heart rate detection</li><li>Oxygen saturation</li><li>30 days continuous battery life</li></ul> |
| **Data Features** | <ul><li>30 days on-device data storage</li><li>Data is downloaded from device via USB</li></ul> | <ul><li>100 days cloud data storage</li><li>Remote data download/Bluetooth</li></ul> Sleep depth estimation (light, deep, or REM) which has been validated against PSG | <ul><li>Extraction mode settings:</li><li>30 days on-device storage via USB download</li><li>100 days cloud data storage via Bluetooth</li><li>Sleep depth estimation (light, deep, or REM) which has been validated against PSG</li></ul> |
| **Features Consistent Across All Devices** | <ul><li>30 s epochs</li><li>Rechargeable battery</li><li>Water-resistant to 50 m</li><li>Dual function as a watch and alarm clock</li><li>Event marker</li><li>Optional feedback mode</li><li>Companion software which allows researchers to score raw activity data</li><li>CSV export of epoch-by-epoch (EBE) data</li><li>CSV export of summary data averaged across all study days</li><li>Bedtime in 24 h clock time</li><li>Waketime in 24 h clock time</li><li>Total sleep time (TST) in minutes</li></ul> Sleep efficiency (SE) as a percentage | | |

## 3. Results

### 3.1. Respondents

#### 3.1.1. Recruitment Exposure

The total estimate of exposure of potential respondents through all platforms was N = 14,057. The vast majority of exposure occurred through the social media website Twitter (N = 11,117). Between April and July 2021, a total of 37 Twitter users clicked the research survey link which had been embedded in recruitment tweets and 14 Twitter users retweeted the recruitment tweets. The estimated exposure through scientific presentations was approximately 142 attendees. Additionally, 101 potential respondents were directly recruited through email.

#### 3.1.2. Demographics

Seventy-six (76) individuals navigated to the Qualtrics survey website. Of these, 55 respondents indicated whether they conducted research related to sleep in real-world environments (see Table 1 above). Nine (N = 9) respondents indicated that they did not

conduct research in this field and were excluded from further participation, leaving a total sample population of N = 46. The respondents did not need to complete the entire survey in order to be included in subsequent analysis. All respondents completed between 5% and 100% of the survey. Sixty-three percent (63%; N = 29) of respondents completed 100% of the rank order survey questions and 39% (N = 18) of respondents completed the HPT.

Figure 1A depicts respondents' years of experience by region: Africa; Asia; Europe; North America; Oceania; and South America; and organization type: Academic or Education Institution; For-Profit Organization or Business; Government; Hospital or Clinical Laboratory; and Non-Profit Organization. The majority of respondents (N = 26) were geographically located in North America and conducted research in an academic or education institution (N = 30). Respondents had an average of 11 years' experience conducting research (Range: 1–27 years; Mode: 10 years). Figure 1B shows the distribution of respondents by research focus across the following categories: (1) healthy sleepers with conventional sleep patterns; (2) sleep disorders; (3) operational environments, such as transportation, shiftwork, or the military; (4) marginalized or underrepresented groups, including racial, ethnic, and gender minorities; and (5) other. One (N = 1) respondent indicated that their research focus pertained to ultra-endurance sports athletes, which constitutes its own category in Figure 1B. Six (N = 6) respondents did not provide an answer. Figure 1C shows the distribution of respondents by research field: (1) biological sciences; (2) behavioral or social sciences; (3) human factors or ergonomics; and (4) diagnostics services. No respondent indicated that they worked in a field other than these categories, and 10 respondents did not provide an answer.

### 3.2. Device Preferences

A qualitative description of responses to Q7–Q12 are depicted in Figure 2. The percentages are shown by total number of respondents per question, rather than the total number of survey respondents. In brief, a 66% majority of respondents (21/32) preferred devices which are worn on the wrist for sleep measurement (Q8) and a 49% majority of respondents (16/33) considered epoch-by-epoch/minute-by-minute to be the most important time scale for measuring sleep (Q8). On Q9, 52% (15/29) believed that the most appropriate method for determining sleep onset/offset was through a combination of brain activity, motor activity (e.g., activity counts), and peripheral biometric measures (e.g., heart rate and/or oxygen saturation). Eighty-three percent (83%; 24/29) of respondents indicated that they collected data related to napping or sleep fragmentation (Q10). Only these 24 respondents were permitted to answer Q11. On Q11, 52% (12/23) indicated that the most appropriate minimum period of inactivity that could be considered a nap was less than or equal to 20 min. Another sizeable fraction (44%) of respondents selected a time period of between 20 and 40 min. No respondents selected a time period greater than 60 min for Q11. For Q12, a 44% majority of respondents (14/32) preferred a continuous observation window of between 4 and 14 days long.

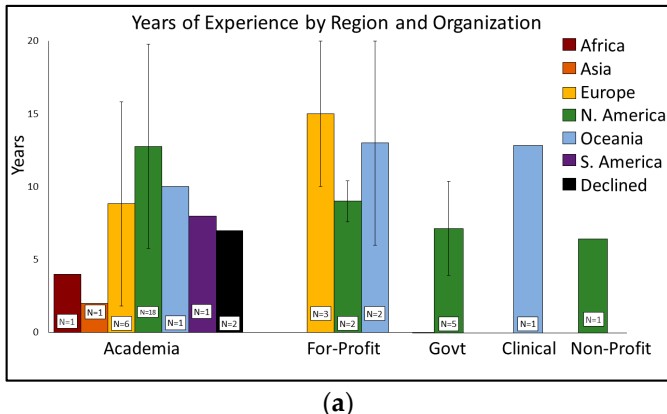

(**a**)

**Figure 1.** *Cont.*

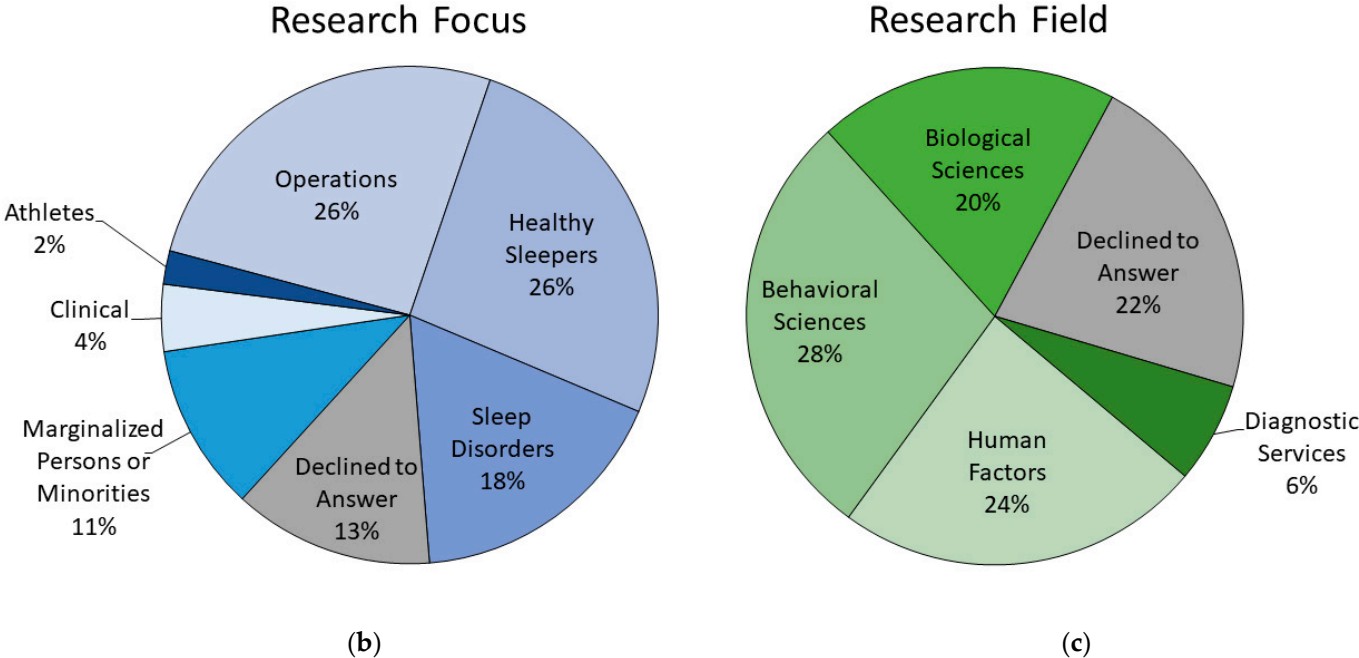

**(b)** **(c)**

**Figure 1. Respondent demographics by years' experience, region, and research:** (**a**) Respondents' years of experience (*y*-axis; mean ± standard deviation) plotted by organization (*x*-axis) and global region (legend). The number of respondents (N) per category is indicated by the white inset box. The percentage of respondents by (**b**) research focus and (**c**) research field are depicted in the pie charts.

Rank order responses are depicted in Figure 3. For Q13, total sleep time (TST) received the highest ranking for information about sleep that respondents considered important to their research, followed by objective sleep quality, time in bed (TIB), and subjective sleep quality. Measures of sleep architecture or sleep depth and diagnostic information (e.g., apnea-hypoxia index, AHI) were ranked the lowest out of the provided categories. One respondent provided a write-in response indicating that social activity timing was of high importance to their research. Regarding device features which facilitated data collection (Q14), the ability to differentiate between actual sleep and "false sleep" (i.e., a period of low activity that mimics sleep onset but is not a sleep episode) was most frequently ranked as having high importance, followed by data security. The ability for subjects to self-report information other than sleep and the capacity to provide feedback or an intervention were ranked the lowest. For Q15, the top-ranked factor which respondents felt limited their observation period or window of data collection was battery life. Logistics, such as receiving approval for study procedures, was ranked the lowest. Four (4) respondents provided additional comments about the use of CSTs for research. The end-of-survey comments are included in Appendix A.

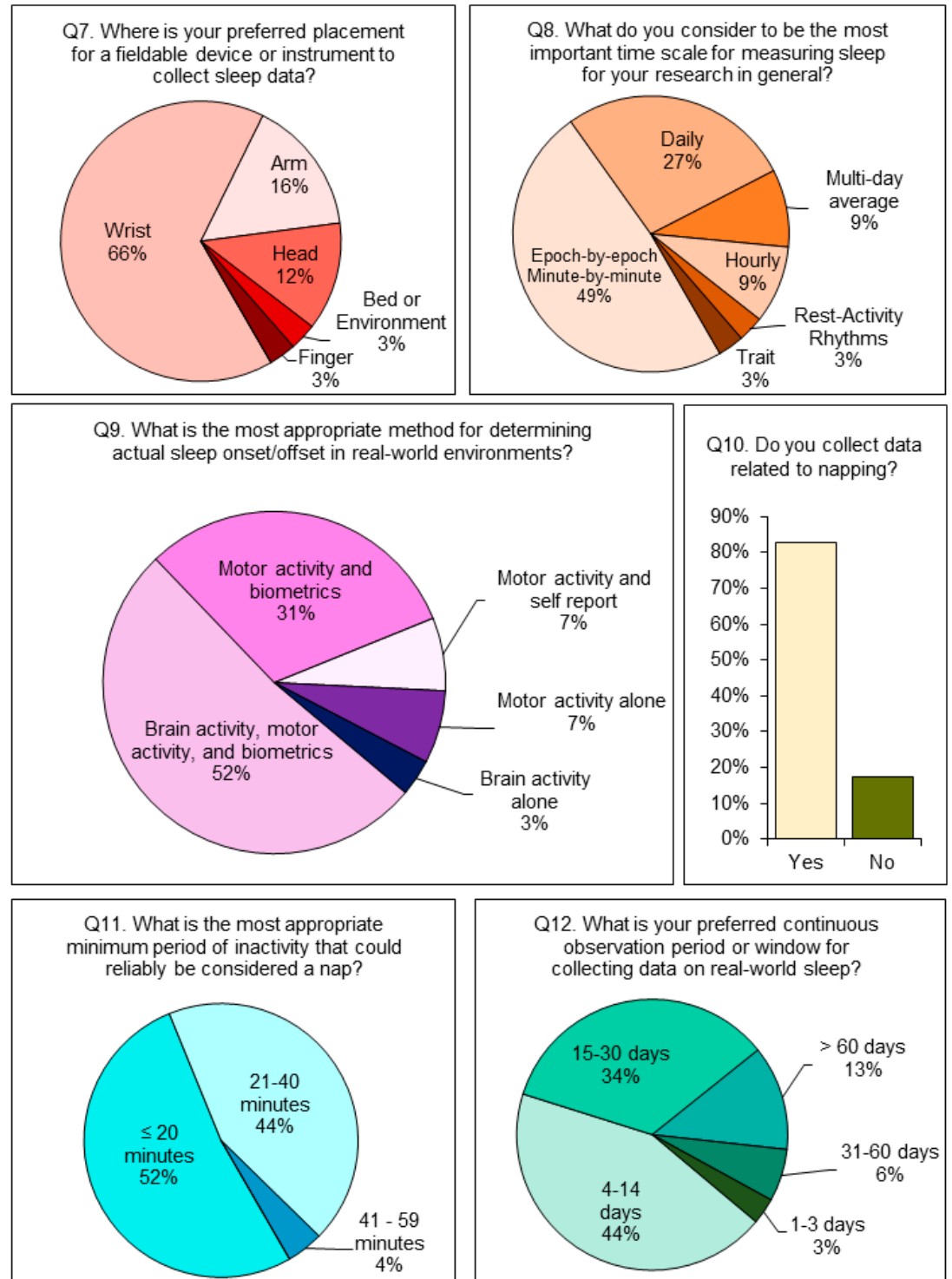

**Figure 2. Device preferences: multiple choice:** Responses to Q7–Q12 regarding device preferences by number of respondents per question. The pie slices indicate multiple-choice options and the percent of respondents per question who selected this response. The multiple choice options which were not selected by any respondent (0%) are not depicted.

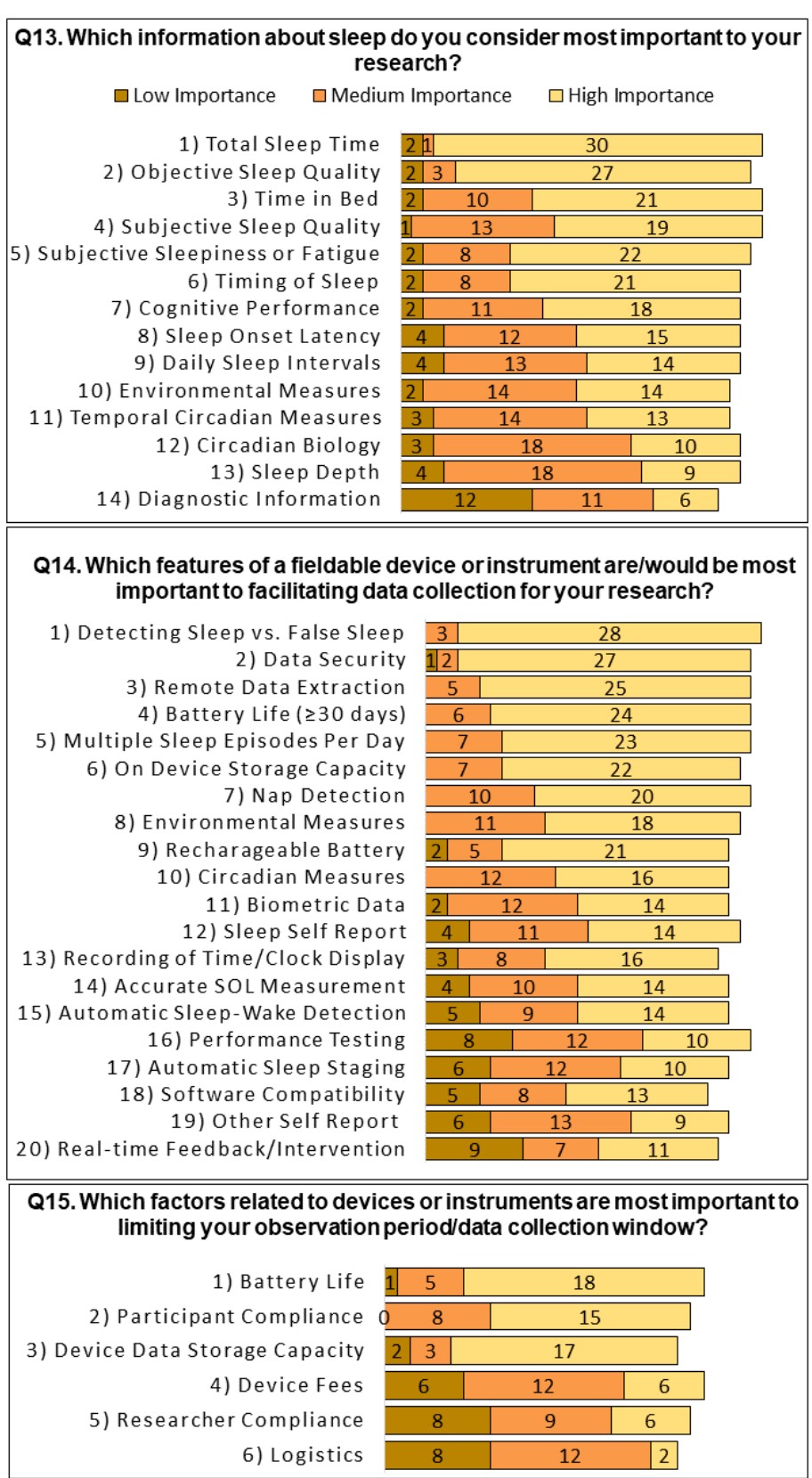

**Figure 3. Device preferences: rank order:** Mean rank order of responses to Q13–Q15 regarding device preferences by level of importance. The items are listed on the *y*-axis by weighted rank, with number 1 corresponding to higher importance rankings. The bars depict the number of responses by level of importance (low importance in dark gray, medium importance in gray, and high importance in light gray) for each item.

*3.3. Behavioral Economic Demand*

Demand curves were created for the hypothetical devices described in Table 2. The demand curves are depicted in Figure 4. The Gilroy et al. (2021) model of demand [33] fit the data well, with an average $R^2$ of 0.9847. Post hoc extra sum of squares F-tests determined a significant difference between the curves for the three devices ($F(2, 447) = 6.01$, $p = 0.003$). Further examination showed that Device B had a significantly lower $Q_0$ than Devices A and C, indicating that when the devices are free, participants would 'purchase' fewer units of Device B than Devices A and C. Additionally, the F-tests on $\alpha$ values indicated that the demand for Device B was significantly more elastic than the demand for Devices A and C. The essential values for Devices A, B, and C were 705, 367, and 935, respectively. In rank order, Device C maintained the highest purchasing level across prices, while Device B maintained the lowest.

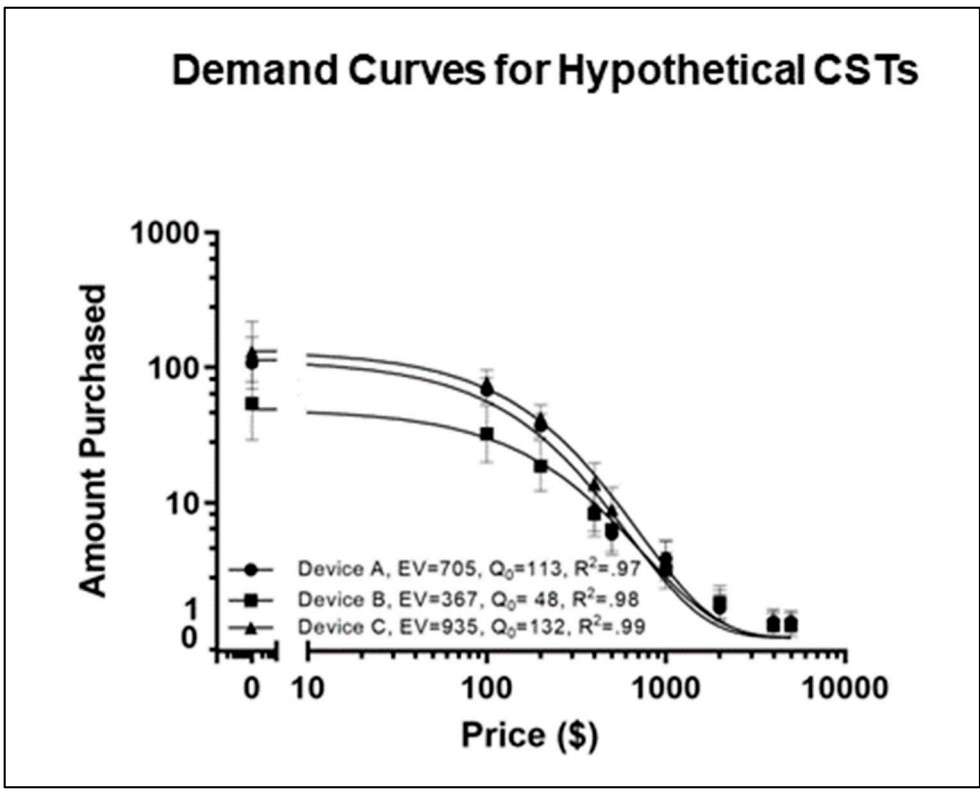

**Figure 4. Demand curve for hypothetical CST devices:** Group mean demand curves for Device A, Device B, and Device C. The essential value (EV) indicates the degree to which consumption level resists the impact of increases in price, $Q_0$ is an estimate of the maximum level of demand, and $R^2$ represents the coefficient of determination. Consumption data is shown in inverse hyperbolic sine (IHS) units, a log-like scale that evaluates at zero.

## 4. Discussion

The main aim of the survey was to establish a reliable consensus opinion from sleep research professionals about the preferred device features for measuring sleep outside the laboratory in the context of economic demand. This is important not only to provide guidance to CST or wearables manufacturers, but also to facilitate innovation within sleep research methodology. The reliability of the survey results hinges on the assumption that a sufficient number of subject matter experts provided responses. Survey completion was done anonymously to ensure that respondents would feel comfortable providing honest responses, but it could create skepticism about the expertise of unknown respondents. While no identifying information was collected, respondents had to indicate that they conducted sleep research related to human sleep in real-world environments in order to complete the survey. Expertise may be considered a matter of opinion; for the purposes of

this survey, the included respondents were those who considered themselves sufficiently knowledgeable to participate. The low number of respondents is a study limitation, and the results must be interpreted as descriptive rather than statistically significant.

Many findings are unsurprising, such as researchers' high ranking of the importance of sleep duration (TST and TIB) and their interest in a device which can reliably differentiate sleep from inactivity and provide data security (Figure 3, Q14). Accurate measurement of sleep is a prerequisite for the use of any device in a research setting [6,34]. Moreover, ensuring privacy and data security is required by institutional review boards (IRB) in order to conduct human research studies. However, the survey highlighted a few areas where CSTs could make improvements. Notably, estimations of sleep depth were considered less important to respondents than almost any other information about sleep (Figure 3, Q14). It may be that researchers' disinterest in sleep depth estimations are related to the fact that the majority of CSTs do not measure sleep architecture such as PSG (N1, N2, SWS, or REM), but instead provide non-equivalent measures (e.g., light, deep, or REM) with little documentation with regards to the scoring criteria [35–37]. It is possible that researchers would be more interested in sleep staging capabilities if CST systems reliably measured PSG-equivalent sleep architecture and followed standardized scoring criteria, or if sleep depth estimations were shown to be robustly related to sleep health outcomes.

While deep sleep may be connoted with superior quality, sleep depth was listed separately from measures of sleep quality in this survey. Objective sleep quality, such as number of awakenings, wake after sleep onset (WASO), or sleep efficiency (SE), was ranked by respondents as the second most important information about sleep, and subjective sleep quality, or the sleeper's personal satisfaction with their sleep, was ranked fourth. Both these measures have independently been shown to be important for health and performance outcomes [38–42], which may explain why they were highly ranked by sleep experts. The value of nap detection and the minimum time period required for sleep determination is indicated by the finding that 83% of respondents collect data related to napping and a combined 96% preferred naps to be defined as a sleep event less than or equal to 40 min (Figure 2, Q10–Q11). The number of times per day sleep occurs (daily sleep intervals; DSI) was ranked ninth in importance (Figure 3, Q13). Nap information is important to researchers working in operational domains such as shiftwork, healthcare, or aviation. In these safety-sensitive industries, strategic napping is recommended to prevent on-the-job fatigue [43–48]. Napping is also important as a health indicator across all research populations [49–51]. The impact of napping on downstream effects cannot be assessed if devices do not record them.

Respondents overwhelmingly indicated that periods of inactivity shorter than an hour could reliably be considered a nap (Figure 2, Q11). No respondents indicated that minimum nap detection should be greater than 60 min. Few studies have assessed the performance of CSTs for measuring naps and the specific parameters regarding minimum duration for scoring sleep are poorly communicated by CST companies [6]. A CST (the Zulu watch) that has been validated against PSG [52] automatically detects sleep episodes as short as 20 min using only accelerometry, indicating that algorithms can be developed to detect short naps. Automatic detection of short sleep periods and multiple sleep periods per day is one area in which developers could improve CSTs with regards to scientific merit.

The importance of battery life was highlighted by a number of survey items. Battery life was ranked as the most important factor which limited researchers' observation window (Q15), and extended battery life (defined as greater than/equal to 30 days) was the fourth highest ranked feature of a device which would facilitate data collection (Q14). Having to recharge or replace the battery frequently can result in periods of missing data during data collection. The majority of respondents (44%) preferred a continuous data collection window of between 4 and 14 days long, based on responses to Q12. Thirty-four percent (34%) preferred 15–30 days as an observation window. As shown in Figure 4, economic value and demand were greater for hypothetical devices with longer battery life on the HPT.

The findings suggest that extending battery life is another area in which CST developers could enhance competitive value in the scientific market.

Respondents' preferences were occasionally contradictory. For example, respondents preferred devices which measure sleep at the wrist (Q7) but believed the most appropriate method for determining sleep onset in real-world environments was by measuring brain activity in combination with motor activity and biometric data (Q9). Brain activity cannot currently be measured at the wrist. The next most popular method of sleep/wake determination, which was selected by 31% of respondents, was by a combination of motor activity and biometrics. This is how many CSTs determine sleep, but it is not the method used by actigraphy [14,53]. Only 7% of respondents preferred a standard actigraphy method for sleep–wake determination, either through motor activity alone or motor activity in combination with self-report (Q9). This finding would suggest that researchers actually prefer the sleep determination method used by CSTs to actigraphy.

Respondents also rated data security and remote data extraction closely with respect to desirable features (Q14). Demand was greatest for a hypothetical device which had the options of either remote or wired data extraction, but it was lowest for a device that only featured remote data extraction. Remote data extraction would likely rely on wireless technology which utilizes an automatic sync function to upload data to a cloud-based or internet-enabled server. Ensuring data security through a system such as this is not impossible, but it presents more areas of vulnerability than a wired data transfer to an offline computer.

The behavioral economic analysis of demand curves for the three hypothetical devices provides insight into decision-making behavior about purchasing CSTs for research. HPTs are typically used to assess individual commodity demand (for example, the demand for alcohol and drugs) and their reliability and validity have been well documented [20,54]. The demand curves created here demonstrated significantly lower and more elastic demand for Device B than the other two devices. The main features that distinguish the device with the lowest demand (Device B) and the device with the highest demand (Device C) were long battery life (4 days for Device B and 30 days for Device C), and the ability to choose between extracted data wirelessly via Bluetooth and through a wired download (USB). A long battery life was indicated as one of the most-desired features for CSTs, and a short battery life was also indicated as the most limiting factor for research studies. Device A featured a longer battery life (30 days, and data that could be extracted only through a USB download. There was significantly more demand for Device A than for Device B, despite the fact that Device B included more features, including sleep depth estimation, which had been validated against PSG. This finding supports the ranking of sleep depth estimation having low importance to sleep researchers from Figure 3. From these data, it is likely that the demand for Devices A and C was largely driven by their longer battery life features and secure wired data extraction features.

The low importance of sleep depth scoring and the high importance of short nap detection, data security, and battery life are at odds with the features of the majority of CST wearables on the market today. IBR is conducting a follow-up analysis to compare the features of currently available devices against expert preferences. The contrast between what scientists consider important and what is being produced suggests a fundamental shortcoming of the market-driven approach to the design of sleep-tracking wearables. The average consumer is not necessarily the best judge of what is important to measure about a physiological process like sleep, despite their interest in the health science aspect of a device. This gap is also an opportunity: it is clear that the current technologies are capable of delivering features that have scientific merit. A company that orients toward these features would have a unique market advantage; perhaps not just within the academic research community, but within the larger consumer populace as well. The next step in this project is to determine whether the average consumer is willing to pay more for a device with scientifically meritorious sleep-tracking features and to measure how scientific validation

or endorsement impacts the hypothetical demand for a sleep monitoring wearable in consumers who indicate an interest in monitoring sleep.

A highlight of the methodology of this survey has been the attempt to recruit opinions from as many sleep researchers as possible from across multiple research domains. Previous publications about scientifically meritorious CST features may reflect the views of the authors or the opinions from a select workshop panel of invited experts [5–8,10–13,34,35,53,55]. These works are of great importance to the field, but exclude the valid contributions from early-career, industry, or non-academic researchers. Moreover, researchers may be more likely to provide their honest, yet unpopular, beliefs about CSTs in an anonymous survey conducted privately rather than through a live discussion or named authorship. Another highlight is that this is the first report of economic demand for hypothetical devices within a sleep researcher population.

A limitation to the interpretation of these results is the low sample size. Because the actual global population of sleep researchers who have expertise in data collection outside the laboratory is unknown, the statistical power of this sample size (N = 46) is indeterminate. Great efforts were taken to recruit as many real-world sleep experts as possible, and a variety of regions and research domains were ultimately represented (Figure 1). However, these findings are limited to a qualitative case study and may not generalize to the overall sleep researcher population. Relatedly, the survey was focused on establishing consensus opinion among sleep researchers most interested in naturalistic "real-world" sleep, rather than medical clinicians. While 18% of respondents indicated that they conducted research related to sleep disorders, researchers may be less interested in diagnostic or treatment features for CSTs. CSTs traditionally have had lower accuracy for the measurement of sleep in disordered populations relative to healthy controls. The current results, therefore, may not generalize to the medical community; a follow-up survey will concentrate on device preferences for diagnostic purposes, rather than for the scientific observation of naturalistic sleep behavior. The survey was conducted in English, which may have led to skewed demographics. The skewed demographics and low sample size of this case study also prevent comparisons between respondents by culture, nationality, or affiliation. Another limitation is that respondents were not evenly distributed across research domains, precluding a comparison between domains and years' experience regarding CST preferences. Respondents had, on average, 11 years' experience conducting sleep research, with 10 years being the most frequently indicated length of experience. It is interesting to note that CSTs have only been around for 10–11 years themselves [1,5]. Respondents were not asked to indicate their age, level of education, or terminal degree, so it is possible that the consensus opinions reflect those of early-career researchers. The survey was purposely designed not to ask these questions under the assumption that non-academic researchers, technicians, and students have a valuable perspective on real-world data collection that should not be discounted because of their demographics or degree status. However, an interesting follow-up study would be to evaluate the impact of researcher status on CST preferences or the economic value of their scientific endorsement.

This investigation of CSTs for sleep research purposes differs with respect to previous analyses because it focuses on the desirability of features to a sleep scientist, rather than focusing on the accuracy or validity of sleep measurement. The importance of validation testing of CSTs has been well established in the literature [6,8–10,14], but even a validated device may not translate to researcher desirability. Four (4) respondents provided additional feedback about CSTs in the comments section at the end (see Appendix A). Each of these comments expresses a concern that the data are somehow invalid—either due to arbitrary units, changing algorithms, or improper recording of sleep. While beyond the scope of these analyses, an interesting follow-up survey would be to assess the importance of trust and transparency in the relationship between sleep researchers and CSTs.

Despite the growing conversation about the viability of CSTs for research, manufacturers may not be interested in increasing scientific accuracy in their devices unless doing so is expected to result in greater consumer sales. While the global sleep tracking device market

is estimated to be worth up to USD 50 billion by 2027 [56], the monetary value of a scientific endorsement of a product or specific, scientifically relevant design features has not been quantified. The next study in our project will address whether increasing the value of a CST product within the sleep research community, backed by independent validation and endorsement, results in an increase in the value of the product for the general consumers interested in purchasing a sleep tracker.

## 5. Conclusions

Consensus opinion from this survey indicates that real-world sleep experts are looking for an accurate and reliable multi-sensor, wrist-worn device that measures sleep on an epoch-by-epoch basis, detects sleep as short as 20 min, and has sufficient battery life to record data continuously for between 4 and 30 days. Real-world researchers are more interested in measures of sleep duration and quality than they are in sleep depth estimation, diagnostic information, circadian measures, or cognitive performance. The device should be able to remotely extract data, but it needs to provide data security, perhaps through a feature which allows researchers to turn off wireless capabilities. The features ranked most highly as important to real-world sleep researchers do not align with the most prominent features of currently marketed wearables. These data provide context for further behavioral economics analyses to determine the differences in demand for scientifically meritorious CST device features between scientists and general consumers in order to inform IoT business and development decisions.

**Author Contributions:** Conceptualization, J.K.D., L.P.S. and S.R.H.; methodology, J.K.D., L.P.S., J.C. and S.R.H.; formal analysis, J.K.D., L.P.S., J.C. and S.R.H.; data curation, J.C.; writing—original draft preparation, J.K.D., L.P.S. and S.R.H.; writing—review and editing, J.K.D., L.P.S., J.C. and S.R.H.; visualization, J.K.D. and L.P.S.; supervision, S.R.H. All authors have read and agreed to the published version of the manuscript.

**Funding:** This research received no external funding.

**Institutional Review Board Statement:** This study was conducted according to the guidelines of the Declaration of Helsinki and approved by Salus Institutional Review Board for the Institutes for Behavior Resources, INC. (Protocol Number IBR2021, March 2021).

**Informed Consent Statement:** Informed consent was obtained from all subjects involved in the study.

**Data Availability Statement:** Data from this survey can be provided upon request.

**Acknowledgments:** The authors would like to acknowledge and thank the survey respondents for providing their time and honest opinions. The authors would also like to thank any individuals who shared information about the survey with their colleagues or the larger sleep research community.

**Conflicts of Interest:** The authors declare no conflict of interest.

## Appendix A. Comments

**Comment 1:** In my experience, the data/feedback on the device and/or accompanying phone app need to be immediately understandable and trusted by the user. I've seen compliance and buy-in drop off quickly when subjects don't feel like they are getting useful or trusted data on a daily or weekly basis. If the device even misses picking up just one or two sleep episodes or is really inaccurate (e.g., detecting sleep onset or offset >1 h from reality, or not picking up a major awakening in the night) can break trust easily, especially in the beginning stages of a study. Then subjects think its [sic] just a junk device. So having a device with immediate buy-in is huge, especially for longer-term studies (>1 week) where subjects aren't necessarily interacting with researchers daily.

**Comment 2:** One of the concerns that we run into with consumer products are the security issues surrounding the pipeline for cloud-based data and the use of a participant's phone (app-based). There is also the issue that was not explicit in the questionnaire as to whether the raw data are available or only the consumer company provided summary

(EBE or whole night). Given that the company algorithms can change (improve!?), this would disrupt a longitudinal study or a study conducted over multiple years.

**Comment 3:** Activity counts are arbitrary/meaningless units; they are undefined. Actigraphy is only reliable for measuring the daily timing of activity (wakefulness?). A defined, unit of measure for "activity count" needs to be developed/established . . . like mph, bpm. Variability in the sensitivity to same motion/movement among the same or different devices is too great. E.g., Place 2 or more actigraphy devices on your wrist (same Make: Model) and record simultaneously (1 min epoch) for a few hours or 24 h. The activity count values for corresponding minutes are not similar . . . not even close. If you don't have a unit of measure, how can you calculate/conclude anything?

**Comment 4:** Both actigraphy and CSTs are proxy measures of sleep and both require significantly more work to have any confidence at all that they can reliably and accurately distinguish sleep and wake.

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
