# Peer review of "Expert Demand for Consumer Sleep Technology Features and Wearable Devices: A Case Study"

_2624-831X, doi:10.3390/iot3020018_

Round 1
Reviewer 1 Report
Major Comments
- The abstract can be rewritten to be more meaningful. The authors should add more details about their results in the abstract. Abstract should clarify what is exactly proposed (the technical contribution) and how the proposed approach is validated.
- What is the motivation of the proposed work?
- Introduction needs to explain the main contributions of the work clearer.
- The novelty of this paper is not clear. The difference between present work and previous Works should be highlighted.
- Authors must explain in detail the introduction section.
- Research gaps, objectives of the proposed work should be clearly justified.
- To improve the Related Work and Introduction sections authors are highly recommended to consider these high-quality research works , < A Novel Adaptive Battery-Aware Algorithm for Data Transmission in IoT-Based Healthcare Applications >
- English must be revised throughout the manuscript.
- Limitations and Highlights of the proposed methods must be addressed properly
Author Response
Reviewer 1
Comment 1: The abstract can be rewritten to be more meaningful. The authors should add more details about their results in the abstract. Abstract should clarify what is exactly proposed (the technical contribution) and how the proposed approach is validated.
Response: The abstract has been rewritten to be more meaningful and clear per the reviewer’s requests. However, the amount of detail requested by the reviewer cannot be accomplished within the confines of the 200 word limit for Abstracts. We have attempted to address these issues throughout in the Introduction section.
Comment 2: What is the motivation of the proposed work? Introduction needs to explain the main contributions of the work clearer. Research gaps, objectives of the proposed work should be clearly justified.
Response: The Introduction has been modified throughout to clarify research gaps and the objectives of the current project. The specific motivation for the project has been added to lines 66-68: “The motivation for this project is to describe the value of scientific meritorious CSTs in quantifiable terms, i.e., United States dollars (USD), that can then be used to guide the development of future CSTs in order to maximize scientific as well as economic gains.” We hope that these edits have clarified the purpose of the study.
Comment 3: The novelty of this paper is not clear. The difference between present work and previous Works should be highlighted. Authors must explain in detail the introduction section.
Response: The novelty of the paper is that the preferences of sleep research experts for devices by features and price has not previously been assessed. The Abstract, Introduction, and Discussion sections have been modified to make this point more clear. Specifically, text has been rewritten on lines 10-14: “CST marketing campaigns often advertise the scientific merit of devices, but these claims may not align with consensus opinion from sleep research experts. Consensus opinion about CST features has not previously been established in a cohort of real-world sleep research experts. This case study reports the results of the first survey of sleep research experts designed to identify CST metrics and features that are most desirable to the scientific community”; on lines 65-70: “This paper presents the results from an expert survey eliciting the professional opinions and economic valuation of CSTs from a sample of real-world sleep researchers. To our knowledge, this is the first assessment of its type, and also the first report in an on-going, multi-step project conducted as a collaboration between the Institutes for Behavior Resources (IBR) Operational Fatigue and Performance group and the Behavioral Economics group to establish the economic value of CST wearable features on the basis of scientific merit”; and on lines 418-424: “Previous publications about scientific meritorious CST features may reflect the views of the authors or the opinions from a select workshop panel of invited experts [5,7-9,11-14,33,34,52,55]. These works are of great importance to the field, but exclude the valid contributions from early-career, industry, or non-academic researchers. Moreover, researchers may be more likely to provide their honest, yet unpopular, beliefs about CSTs in an anonymous survey conducted privately rather than through a live discussion or named authorship.”
Comment 4: To improve the Related Work and Introduction sections authors are highly recommended to consider these high-quality research works , < A Novel Adaptive Battery-Aware Algorithm for Data Transmission in IoT-Based Healthcare Applications >
Response: The reviewer’s suggested papers are relevant to advancements being made in the field of consumer sleep tracking, and particularly battery life. The authors could not find a way to integrate these citations into the current manuscript, which is focused on hypothetical demand, but we will be sure to reference this work as applicable in future project reports. Thank you for bringing these works to our attention.
Comment 5: English must be revised throughout the manuscript.
Response: The English in the manuscript has been reviewed by the full author team. All authors are native English speakers from the United States. We apologize if any phrasing, spelling, or colloquialisms follow American conventions that may be different from language standards used in other English-speaking regions.
Comment 6: Limitations and Highlights of the proposed methods must be addressed properly
Response: The limitations and highlights of the methods have been discussed at length in the Discussion section on lines 416-449: “A highlight of the methodology of this survey has been the attempt to recruit opinions from as many sleep researchers as possible from across multiple research domains. Previous publications about scientific meritorious CST features may reflect the views of the authors or the opinions from a select workshop panel of invited experts [5,7-9,11-14,33,34,52,55]. These works are of great importance to the field, but exclude the valid contributions from early-career, industry, or non-academic researchers. Moreover, researchers may be more likely to provide their honest, yet unpopular, beliefs about CSTs in an anonymous survey conducted privately rather than through a live discussion or named authorship. Another highlight is that this is the first report of economic demand for hypothetical devices within a sleep researcher population.
A limitation to the interpretation of these results is the low sample size. Because the actual global population of sleep researchers who have expertise in data collection outside the laboratory is unknown, the statistical power of this sample size (N=46) is indeterminate. Great efforts were taken to recruit as many real-world sleep experts as possible, and a variety of regions and research domains were ultimately represented (Figure 1). However, these findings are limited to a qualitative case study and may not generalize to the overall sleep researcher population. Relatedly, the survey was focused on establishing consensus opinion among sleep researchers rather than medical clinicians. The current results may not generalize to the medical community; a follow-up survey will concentrate on devices preferences for diagnostic purposes rather than scientific observation of naturalistic sleep behavior. The survey was conducted in English, which may have led to skewed demographics. Another limitation is that respondents were not evenly distributed across research domains, precluding a comparison between domains or years’ experience regarding CST preferences. Respondents had, on average, 11 years’ experience conducting sleep research, with 10 years being the most frequently-indicated length of experience. It is interesting to note that CSTs have only been around for 10-11 years themselves [1,5]. Respondents were not asked to indicate their age, level of education, or terminal degree, so it is possible that the consensus opinions reflect those of early-career researchers. The survey was purposely designed not to ask these questions under the assumption that non-academic researchers, technicians, and students have a valuable perspective on real-world data collection that should not be discounted because of their demographics or degree status. However, an interesting follow-up study would be to evaluate the impact of researcher status on CST preferences or the economic value of their scientific endorsement.”
Reviewer 2 Report
The authors present an excellent case study of the demand for consumer sleep technology features and wearable devices. The procedure of collecting data is clearly described. The analysis of the data received from the survey is deep. Therefore, I tend to accept the paper.
Page 5: Specifications and Data Features are the same. Are they redundant or typos?
Author Response
The authors present an excellent case study of the demand for consumer sleep technology features and wearable devices. The procedure of collecting data is clearly described. The analysis of the data received from the survey is deep. Therefore, I tend to accept the paper.
Response: Thank you very much for your kind review and for taking the time to thoroughly evaluate this manuscript.
Comment 1: Page 5: Specifications and Data Features are the same. Are they redundant or typos?
Response: This typo has been fixed in Table 2.
Reviewer 3 Report
General Comments
The aim of the work is to answer three main questions related to Consumer Sleep Technologies (CSTs) for providing information about the most important features that CSTs should have from a scientific perspective. Indeed, Experts on sleep quality analysis have been recruited on the basis of their scientific profile.
The work is well presented and may be considered as a first step for designing wearable devices, capable of providing more accurate estimates on sleep quality.
I believe that this work can be accepted to MDPI IoT journal as a contributing paper.
Specific Comments
Few minor comments:
- Table 2: Specifications and Data Features have the same items. Why don't you use a single line?
- Please uniform the text justification.
Author Response
The aim of the work is to answer three main questions related to Consumer Sleep Technologies (CSTs) for providing information about the most important features that CSTs should have from a scientific perspective. Indeed, Experts on sleep quality analysis have been recruited on the basis of their scientific profile. The work is well presented and may be considered as a first step for designing wearable devices, capable of providing more accurate estimates on sleep quality. I believe that this work can be accepted to MDPI IoT journal as a contributing paper.
Response: Thank you very much for your kind review and for taking the time to thoroughly evaluate this manuscript. The authors appreciate your insight about the need to address shortcomings in currently-available wearable devices.
Comment 1: Table 2: Specifications and Data Features have the same items. Why don't you use a single line?
Response: This typo has been fixed in Table 2.
Reviewer 4 Report
The major problems of your study are bias and interpreting the results.
- Selection Bias. Your target population is sleep experts but only one expert in your study is from hospital, although 18% of respondent working wit sleep disorders. A major interest of using wearable devices like smartwatch by the general public is that people care about their sleep health. Most people do not care about the sleep duration measured by smartwatch, because they usually know when they go to bed and when they wake up, or they can track their sleep using sleep log. People use smartwatch usually because they may suspect that they have some sleep disorders thus they want to measure their sleep using wearables. Some common sleep disorders such as sleep apnea and insomnia which have a prevalence even up to 30-40% in the general population can cause huge healthcare burden, for example in US the undiagnosed sleep apnea cost 150 billion USD per year (https://aasm.org/economic-burden-of-undiagnosed-sleep-apnea-in-u-s-is-nearly-150b-per-year/). The voice of sleep experts with background in sleep medicine in this study is extremely low. In fact in basic human sleep research a considerable number of experts are actually working in the field of circadian rhythm and chronobiology because sleep is a process of combining sleep regulation and circadian rhythm (i.e., the well-known S&C model), and these experts seem also to be missing in your study because of the low rank of circadian biology in your results. These selection biases may mislead industry CST developers.
- Survival Bias. The selection of survey authors can lead to survival bias. You selected authors who had published at least two papers using CST. Currently, most CST devices were only validated in measuring sleep-wake, i.e., total sleep time (TST) in sleep research. Thus, scientists who have published studies using CST are likely to study changes in TST in their studies. If you ask them which parameters are most important in their research, of course TST is likely to be ranked high. You should ask the respondents what sleep parameters did they measure in their study. I also worked with CST in measuring sleep in health controls and patients, which required technical support from the CST producers because they rarely provide raw data measured by their devices to customers. My experience is many sleep experts are willing to use CST in their studies, but they cannot do that because of technical issues, such as handling big data measured by CST, access to raw data, and the accuracy of measurement cannot satisfy their study objective. Those sleep experts cannot publish at least 2 papers, but they are the majority of the sleep society and they may have totally different demands for CST (i.e., their demands for CST are usually beyond the simple TST measurements but cannot be met by current products).
- In your results of Q8, about half of the experts consider the epoch-by-epoch sleep is the most important. Why do they want epoch-by-epoch sleep? I think the authors misunderstand TST and epoch-by-epoch sleep. In sleep research we measure epoch-by-epoch sleep usually because we want to measure sleep stages and sleep architecture. Total sleep time is not the purpose of epoch-by-epoch sleep analysis.
- The introduction needs to be essentially shortened and improved.
Author Response
Comment 1: The major problems of your study are bias and interpreting the results. Selection Bias. Your target population is sleep experts but only one expert in your study is from hospital, although 18% of respondent working wit sleep disorders. A major interest of using wearable devices like smartwatch by the general public is that people care about their sleep health. Most people do not care about the sleep duration measured by smartwatch, because they usually know when they go to bed and when they wake up, or they can track their sleep using sleep log. People use smartwatch usually because they may suspect that they have some sleep disorders thus they want to measure their sleep using wearables. Some common sleep disorders such as sleep apnea and insomnia which have a prevalence even up to 30-40% in the general population can cause huge healthcare burden, for example in US the undiagnosed sleep apnea cost 150 billion USD per year (https://aasm.org/economic-burden-of-undiagnosed-sleep-apnea-in-u-s-is-nearly-150b-per-year/). The voice of sleep experts with background in sleep medicine in this study is extremely low. In fact in basic human sleep research a considerable number of experts are actually working in the field of circadian rhythm and chronobiology because sleep is a process of combining sleep regulation and circadian rhythm (i.e., the well-known S&C model), and these experts seem also to be missing in your study because of the low rank of circadian biology in your results. These selection biases may mislead industry CST developers.
Response: The reviewer has made a great observation. The purpose of this survey has been to establish consensus among sleep researchers, not medical clinicians. A large part of the research community focuses on studying sleep disorders, but they do not treat patients. We have added this as a limitation and follow-up project in the Discussion section on lines 431-435: “…the survey was focused on establishing consensus opinion among sleep researchers rather than medical clinicians. The current results may not generalize to the medical community; a follow-up survey will concentrate on devices preferences for diagnostic purposes rather than scientific observation of naturalistic sleep behavior.” Furthermore, we appreciate the reviewer’s insight about the utility of sleep tracking features. We hope that the reviewer will be able to participate in our future surveys, the goal of which is to collect as many expert opinions as possible in a publishable format.
Comment 2: Survival Bias. The selection of survey authors can lead to survival bias. You selected authors who had published at least two papers using CST. Currently, most CST devices were only validated in measuring sleep-wake, i.e., total sleep time (TST) in sleep research. Thus, scientists who have published studies using CST are likely to study changes in TST in their studies. If you ask them which parameters are most important in their research, of course TST is likely to be ranked high. You should ask the respondents what sleep parameters did they measure in their study. I also worked with CST in measuring sleep in health controls and patients, which required technical support from the CST producers because they rarely provide raw data measured by their devices to customers. My experience is many sleep experts are willing to use CST in their studies, but they cannot do that because of technical issues, such as handling big data measured by CST, access to raw data, and the accuracy of measurement cannot satisfy their study objective. Those sleep experts cannot publish at least 2 papers, but they are the majority of the sleep society and they may have totally different demands for CST (i.e., their demands for CST are usually beyond the simple TST measurements but cannot be met by current products).
Response: We appreciate the reviewer’s insight about the difficulties of working with CST data. This kind of real-life experience working with sleep-tracking devices has been the goal of conducting this survey and the project in general. We hope that the reviewer will be able to provide these insights in a future survey. However, the selection of survey respondents was not limited to direct contact following a literature review. Respondents were also recruited through social media, snowballing/word-of-mouth, a letter to the editor in the SLEEP journal, and scientific presentations at the SLEEP conference and National Safety Counsel. The low number of respondents for this survey has been discussed as a limitation for this manuscript. It is our hope that publication of these results will increase visibility of this project and allow IBR to collect more robust data in the future.
Comment 3: In your results of Q8, about half of the experts consider the epoch-by-epoch sleep is the most important. Why do they want epoch-by-epoch sleep? I think the authors misunderstand TST and epoch-by-epoch sleep. In sleep research we measure epoch-by-epoch sleep usually because we want to measure sleep stages and sleep architecture. Total sleep time is not the purpose of epoch-by-epoch sleep analysis.
Response: In Q8, respondents were asked to respond what time scale they considered most important for measuring sleep in their research. The question does not refer specifically to TST nor were respondents asked to indicate best time scale for the measurement of TST. It is possible that respondents prefer to determine TST directly from epoch-by-epoch data by summing the number of epochs that are indicated as sleep against those that are coded as wake, but the authors have not interpreted the results from Q8 to indicate a preference for measuring TST by epoch-by-epoch data in the manuscript.
Comment 4: The introduction needs to be essentially shortened and improved.
Response: The authors have edited the Introduction throughout in order to address the concerns from all four reviewers while keeping the text as concise as possible.
Round 2
Reviewer 1 Report
- Related work section is missing, so can be added
- Introduction section can be rewritten with clear motivation, and significance and novelty
- Authors are highly suggested to add the section/sub-section about the Battery lifetime and Power Mangaement of wearable devices by following these articles ,
- Figs 1 (a)&(b) and 4can be redrawn with high visibility and quality due to the poor and blurrdness
- Authors must clearly justify how their conducted case study is worth demanding? which is unclear from current version
Author Response
Comment 1: Related work section is missing, so can be added. Introduction section can be rewritten with clear motivation, and significance and novelty
Response: The introduction section has been revised throughout to include explicit statements about motivation, related works, significance and novelty of the current project. A stand-alone section on related work was not added in order to follow IoT manuscript formatting guidelines: (https://www.mdpi.com/journal/IoT/instructions).
Comment 2: Authors are highly suggested to add the section/sub-section about the Battery lifetime and Power Mangaement of wearable devices by following these articles ,
Response: The reviewer’s suggested papers are relevant to advancements being made in the field of consumer sleep tracking, and particularly battery life. The authors could not find a way to integrate these citations into the current manuscript, which is focused on hypothetical demand, but we will be sure to reference this work as applicable in future project reports. Thank you for bringing these works to our attention.
Comment 3: Figs 1 (a)&(b) and 4can be redrawn with high visibility and quality due to the poor and blurrdness
Response: Figures 1 and 4 have been redrawn to have higher resolution and less blurriness.
Comment 4: Authors must clearly justify how their conducted case study is worth demanding? which is unclear from current version
Response: The authors apologize for not being able to respond to this comment. Perhaps if the reviewer could rephrase the question, we would be able to answer. Thank you.
Reviewer 4 Report
I accept your explanation that the missing of sleep clinicians' opinions is a limitation of your study. However, I suggest to explain more about the limitation regarding sleep medicine, because some recent studies have shown critical concerns about the accuracy of wearable devices in measuring sleep disorders or some key clinical parameters (e.g., please refer to doi: 10.2196/24171, doi: 10.12688/f1000research.13010.1 , DOI: 10.3390/clockssleep2030027,doi:10.1016/S2213-2600(22)00103-5) in real-world clinical or epidemiological studies. Since these limitations of CST in real world clinical study have been reported, your discussion should include them in the study limitation section. This is much better than simply saying 'future study to do another survey to clinicians', if your study aim to provide useful information for CST developers. I believe that good validation work and good accuracy generally are the main limitations of current CST market in both sleep research and sleep medicine, just like your respondents said in Appendix comments 1,3 and 4 .
The purpose of epoch-by-epoch sleep is not only to measure total sleep time, but also sleep stages, indicating that sleep experts may also prefer to measure sleep stages from their subjects at home in real world studies. This should be mentioned in the discussion.
Nap detection is very important for both circadian rhythm studies (e.g., studies on shift workers) and sleep deprivation studies (e.g., people may take daytime nap to fight against insufficient nocturnal sleep during working days). In addition, nap is also a culture in some countries, such as in China (please refer to doi: 10.2196/13482.). Therefore, the demand of measuring nap may be different between your respondents from different countries. This should be discussed or mentioned as a limitation if you have few respondents. Your further design of survey may need to take this into account as well, to balance the number of sleep experts from different countries or culture background.
Author Response
Comment 1: I accept your explanation that the missing of sleep clinicians' opinions is a limitation of your study. However, I suggest to explain more about the limitation regarding sleep medicine, because some recent studies have shown critical concerns about the accuracy of wearable devices in measuring sleep disorders or some key clinical parameters (e.g., please refer to doi: 10.2196/24171, doi: 10.12688/f1000research.13010.1 , DOI: 10.3390/clockssleep2030027,doi:10.1016/S2213-2600(22)00103-5) in real-world clinical or epidemiological studies. Since these limitations of CST in real world clinical study have been reported, your discussion should include them in the study limitation section. This is much better than simply saying 'future study to do another survey to clinicians', if your study aim to provide useful information for CST developers. I believe that good validation work and good accuracy generally are the main limitations of current CST market in both sleep research and sleep medicine, just like your respondents said in Appendix comments 1,3 and 4 . Nap detection is very important for both circadian rhythm studies (e.g., studies on shift workers) and sleep deprivation studies (e.g., people may take daytime nap to fight against insufficient nocturnal sleep during working days). In addition, nap is also a culture in some countries, such as in China (please refer to doi: 10.2196/13482.). Therefore, the demand of measuring nap may be different between your respondents from different countries. This should be discussed or mentioned as a limitation if you have few respondents. Your further design of survey may need to take this into account as well, to balance the number of sleep experts from different countries or culture background.
Response: These limitations have been added to the Discussion section on lines 420-428: “While 18% of respondents indicated that they conducted research related to sleep disorders, researchers may be less interested in diagnostic or treatment features for CSTs. CSTs traditionally have had lower accuracy for the measurement of sleep in disordered populations relative to healthy controls. The current results, therefore, may not generalize to the medical community; a follow-up survey will concentrate on devices preferences for diagnostic purposes rather than scientific observation of naturalistic sleep behavior. The survey was conducted in English, which may have led to skewed demographics. The skewed demographics and low sample size of this case study also prevents comparisons between respondents by culture, nationality, or affiliation.”
Comment 2: The purpose of epoch-by-epoch sleep is not only to measure total sleep time, but also sleep stages, indicating that sleep experts may also prefer to measure sleep stages from their subjects at home in real world studies. This should be mentioned in the discussion.
Response: This is an interesting interpretation of the data. However, the survey did not collect enough responses on this topic in order to support this possibility as a discussion point. Considering that most CSTs do not collect any EEG information, it could be impossible to stage sleep from epoch-by-epoch CST data. Future versions of this survey will take into consideration respondents’ preferences for scoring raw data compared to automatic scoring by device algorithms, and the granularity of data for purposes of hand scoring and sleep staging. Thank you for bringing this to our attention.